

# CircCamsap1 is dispensable for male fertility in mice

Shu Zhang[1,*], Haojie Li[1,2,3,*], Wei Jiang[4], Xia Chen[1], Han Zhou[1], Chang Wang[5], Hao Kong[3], Yichao Shi[1] and Xiaodan Shi[4]

[1] Center of Reproduction, The Affiliated Changzhou Second People's Hospital of Nanjing Medical University, Changzhou, Jiangsu, China
[2] Changzhou Medical Center, The Affiliated Changzhou Second People's Hospital of Nanjing Medical University, Changzhou, Jiangsu, China
[3] State Key Laboratory of Reproductive Medicine and Offspring Health, Department of Histology and Embryology, Nanjing Medical University, Nanjing, Jiangsu, China
[4] Women's Hospital of Nanjing Medical University, Nanjing Women and Children's Healthcare Hospital, Nanjing, Jiangsu, China
[5] Department of Clinical Nursing, School of Nursing, Anhui University of Chinese Medicine, Hefei, Anhui, China
* These authors contributed equally to this work.

Corresponding authors
Yichao Shi, sina365@163.com
Xiaodan Shi, shi_xiaodan@126.com

## ABSTRACT

**Background:** Circular RNAs (circRNAs) are a large class of RNAs present in mammals. Among these, *circCamsap1* is a well-acknowledged circRNA with significant implications, particularly in the development and progression of diverse tumors. However, the potential consequences of *circCamsap1* depletion *in vivo* on male reproduction are yet to be thoroughly investigated.
**Methods:** The presence of *circCamsap1* in the mouse testes was confirmed, and gene expression analysis was performed using reverse transcription quantitative polymerase chain reaction. *CircCamsap1* knockout mice were generated utilizing the CRISPR/Cas9 system. Phenotypic analysis of both the testes and epididymis was conducted using histological and immunofluorescence staining. Additionally, fertility and sperm motility were assessed.
**Results:** Here, we successfully established a *circCamsap1* knockout mouse model without affecting the expression of parental gene. Surprisingly, male mice lacking *circCamsap1* (*circCamsap1$^{-/-}$*) exhibited normal fertility, with no discernible differences in testicular and epididymal histology, spermatogenesis, sperm counts or sperm motility compared to *circCamsap1$^{+/+}$* mice. These findings suggest that *circCamsap1* may not play an essential role in physiological spermatogenesis. Nonetheless, this result also underscores the complexity of circRNA function in male reproductive biology. Therefore, further research is necessary to elucidate the precise roles of other circRNAs in regulating male fertility.

## INTRODUCTION

Infertility is a condition that impacts millions of individuals and couples worldwide, with the World Health Organization (WHO) estimating that male infertility may account for 50% of infertile couples (*Cox et al., 2022*; *Barratt et al., 2017*). Spermatogenesis is a

precisely regulated, orderly, and complex process. In the seminiferous epithelium, spermatogonial stem cells undergo self-renewal and differentiation, gradually developing into primary spermatocytes. These primary spermatocytes undergo two rounds of meiotic division, ultimately transforming into haploid spermatids. Subsequently, they go through multiple stages of maturation, known as spermiogenesis, ultimately resulting in the formation of mature spermatozoa (*Hess & Renato, 2008*; *Fayomi & Orwig, 2018*; *Ibtisham & Honaramooz, 2020*). Spermatogenesis is regulated by various factors, among which RNA is indispensable in the production of functional sperm (*Morgan et al., 2021*).

Advancement in technologies like high-throughput sequencing, have revealed intricate splicing variants such as circular RNA (circRNA) in complex tissues, notably the testis (*Barbosa-Morais et al., 2012*; *Mele et al., 2015*). Formed through back-splicing, circRNA possesses a closed covalent loop structure. Due to its absence of a cap structure and poly (A) tail, circRNA can elude degradation by exonucleases, ensuring its stability (*Memczak et al., 2013*; *Zhang et al., 2014*; *Chen & Yang, 2015*). Recent studies have illuminated the pivotal roles circRNA plays in diverse biological processes, with implications for male reproduction. For instance, certain circRNAs actively contribute to the self-renewal and differentiation of spermatogonial stem cells (*Li, Ao & Wu, 2018*). Altered expression profiles of circRNAs have been identified in patients with non-obstructive azoospermia (*Ge et al., 2019*). CircSRY, derived from the sex-determining region Y (Sry), a testis-specific gene, serves as a sponge for miR-138 (*Hansen et al., 2013*). The protein encoded by circRsrc1 demonstrates the ability to regulate the assembly of the mitochondrial ribosome during spermatogenesis, consequently impacting male fertility (*Zhang et al., 2023*). While a considerable number of circRNAs have been unearthed in the testis, their functions remain largely elusive (*Wu et al., 2019*; *Tang et al., 2020*). So far, only a small number of circRNAs have been studied for their functions in male reproduction, the functional study of circRNAs still faces significant challenges. One of the reasons lies in how to knock out circRNAs without affecting the expression of their parental genes, and how to avoid the influence of parental genes to only study how circRNAs themselves regulate biological processes (*Yang, Wilusz & Chen, 2022*).

CircCamsap1 is a prominent circular RNA known for its regulatory role in the development and progression of various cancers. Research findings suggest that it can function as a microRNA sponge, influencing the progression of osteosarcoma, colorectal cancer, and hepatocellular carcinoma (*Luo et al., 2021*; *Chen et al., 2021*; *Zhou et al., 2020*). Additionally, studies indicate a potential association between circCamsap1 expression and the status of type 2 diabetes, suggesting its potential utility as a biomarker for this metabolic condition (*Haque et al., 2020*). Camsap1 (Calmodulin Regulated Spectrin Associated Protein 1) and circCamsap1 are both highly expressed in the mouse testes. Meanwhile, studies have found that knockout of Camsap1 results in male infertility (*Hu et al., 2023*). However, the role and regulatory mechanism of circCamsap1 in spermatogenesis remain unclear. In this study, we utilized CRISPR/Cas9 technology to generate circCamsap1 knockout mice. Through this approach, our objective is to elucidate the physiological significance of circCamsap1 in the intricate process of spermatogenesis.

## MATERIALS AND METHODS

### Animal experiments

Protocols for breeding and experiments involving C57BL/6 mice adhered to the guidelines established by the Institutional Animal Care and Use Committee of Nanjing Medical University (Approval No. IACUC-2307030). All mice used in the present study were maintained under a specific pathogen-free (SPF) environment, provided free access to water and food, and were exposed to a 12 h light/dark cycle, a humidity level of 50–55% and a temperature of 23 ± 2 °C. All mice received humane treatment, and every possible measure was taken to reduce any potential distress. When deemed necessary, tissue samples for subsequent analyses were obtained from all mice through euthanasia *via* cervical dislocation. At the conclusion of the study, there were no surviving animals.

To generate *circCamsap1*$^{-/-}$ mice, we designed guide gRNAs targeting the intron between exon 3 and exon 4 of the *Camsap1* gene to knock out the *circCamsap1* rather than the parental gene. After *in vitro* transcription, both Cas9 mRNA and gRNAs were injected into fertilized eggs. Mouse genotypes were identified through PCR amplification, followed by Sanger sequencing for validation. The primer sequences can be found in Table S1.

### RNA extraction, RT-PCR and RT-qPCR

Total RNA was extracted from *circCamsap1*$^{+/+}$ ($n = 3$) and *circCamsap1*$^{-/-}$ mice ($n = 3$) indicated samples by Trizol reagent (15596026; Thermo Fisher Scientific, Waltham, MA, USA). PrimeScript$^{TM}$ RT reagent Kit (RR037A; Takara, Tokyo, Japan) was used for reverse transcription according to manufacturer protocols. Then, we used the cDNA to perform RT-PCR by 2 × Rapid Taq Master Mix (P515; Vazyme, Nanjing, China) or to perform RT-qPCR by AceQ qPCR SYBR Green Master Mix (Q131; Vazyme, Nanjing, China). It was performed using QuantStudio 5 (A28573; Applied Biosystems, Waltham, MA, USA) at 95 °C for 600 s, 95 °C for 10 s, 65 °C for 60 s, 97 °C for 1 s, and 37 °C for 30 s, for a total of 40 cycles. The $2^{-\Delta\Delta CT}$ method was used for calculating the relative transcription level of the target gene. The sequences of primers can be found in Table S1.

### Fertility test

Each adult (8–10 weeks old) male mouse, whether *circCamsap1*$^{+/+}$ or *circCamsap1*$^{-/-}$, was individually mated with three different adult (postnatal day 56) *circCamsap1*$^{+/+}$ C57/BL6 female mice. Fertility assessments were performed by pairing one male mouse with one female mouse over a single night, totaling approximately 12 h. Vaginal plugs were checked, and the number of pups born to each mating pair was recorded to evaluate fertility outcomes. We repeated the experiment three times, with three male mice used for each genotype.

### Histological analyses

Collect testicular and epididymal tissues from a minimum of three *circCamsap1*$^{+/+}$ or *circCamsap1*$^{-/-}$ mice. Place the collected testicular or epididymal tissues in modified Davidson's fluid for 24 h, followed by preservation in 70% ethanol. Proceed with gradient ethanol dehydration, paraffin embedding, and preparation of 5 μm thick slices.
Subsequently, the slices undergo deparaffinization and rehydration for histological assessment using Hematoxylin and Eosin (HE) staining. Spread the sperm on slides, fix at room temperature in 4% PFA for 30 min, and then proceed with HE staining.

## Immunofluorescence

Spermatocyte surface spreading was carried out based on a previous study (*Hua et al., 2019*). The utilized primary antibodies included anti-SYCP3 (diluted 1:200, ab97672; Abcam, Cambridge, UK) and anti-γH2AX (diluted 1:200, ab2893; Abcam, Cambridge, UK). The experiments were repeated three times. Images were captured under the Zeiss *LSM800* confocal microscope.

## Sperm counting and motility assay

Separate the epididymis from adult (postnatal day 56) mice ($n = 3$), and transfer them to modified HTF medium (90126; Irvine Scientific, Santa Ana, CA, USA) containing 10% fetal bovine serum. Open the cauda with a scalpel to release the sperm, and incubate them for 10 min at 37 °C. Subsequently, quantify sperm numbers and motility parameters using computer-assisted semen analysis (Hamilton Thorne Research Inc., Beverly, MA, USA). Evaluate the motility of at least 200 cells for each sample.

## Statistical analysis

All results are presented as the mean ± SEM. GraphPad Prism 7 software (GraphPad Software, San Diego, CA, USA) was used to analyze data and draw graphs. The two-tailed unpaired student's *t*-test was used to compare the differences with the controls. All experiments were repeated at least three times. Statistical significance was accepted if $p < 0.05$. ns, not significant.

## RESULTS

### Expression analysis of *circCamsap1* in mice

As observed in the circBase database, *circCamsap1* is generated through the back-splicing junction between exons 2 and 3 of the *Camsp1* gene (Fig. 1A). To assess the potential impact of *circCamsap1* on spermatogenesis, we initially confirmed its presence in mouse testis. As depicted in Fig. 1B, reverse transcription-polymerase chain reaction (RT-PCR) and Sanger sequencing verified the back-splicing sites of *circCamsap1* in mouse testis. Subsequently, we conducted a multi-tissue expression analysis to explore the expression patterns of both *circCamsap1* and its parent gene, *Camsap1*, in mice. The results revealed high expression levels of both *Camsap1* and *circCamsap1* in mouse testis (Figs. 1C and 1D). For a more detailed examination of the expression dynamics during spermatogenesis, reverse transcription quantitative polymerase chain reaction (RT-qPCR) was employed to quantify transcript levels in mouse testis at different developmental stages. Consistent with the pattern observed in the parental gene *Camsap1*, the expression of *circCamsap1* in mouse testis significantly increased at 21 days after birth, with both continuing to rise gradually until adulthood (Figs. 1E and 1F).

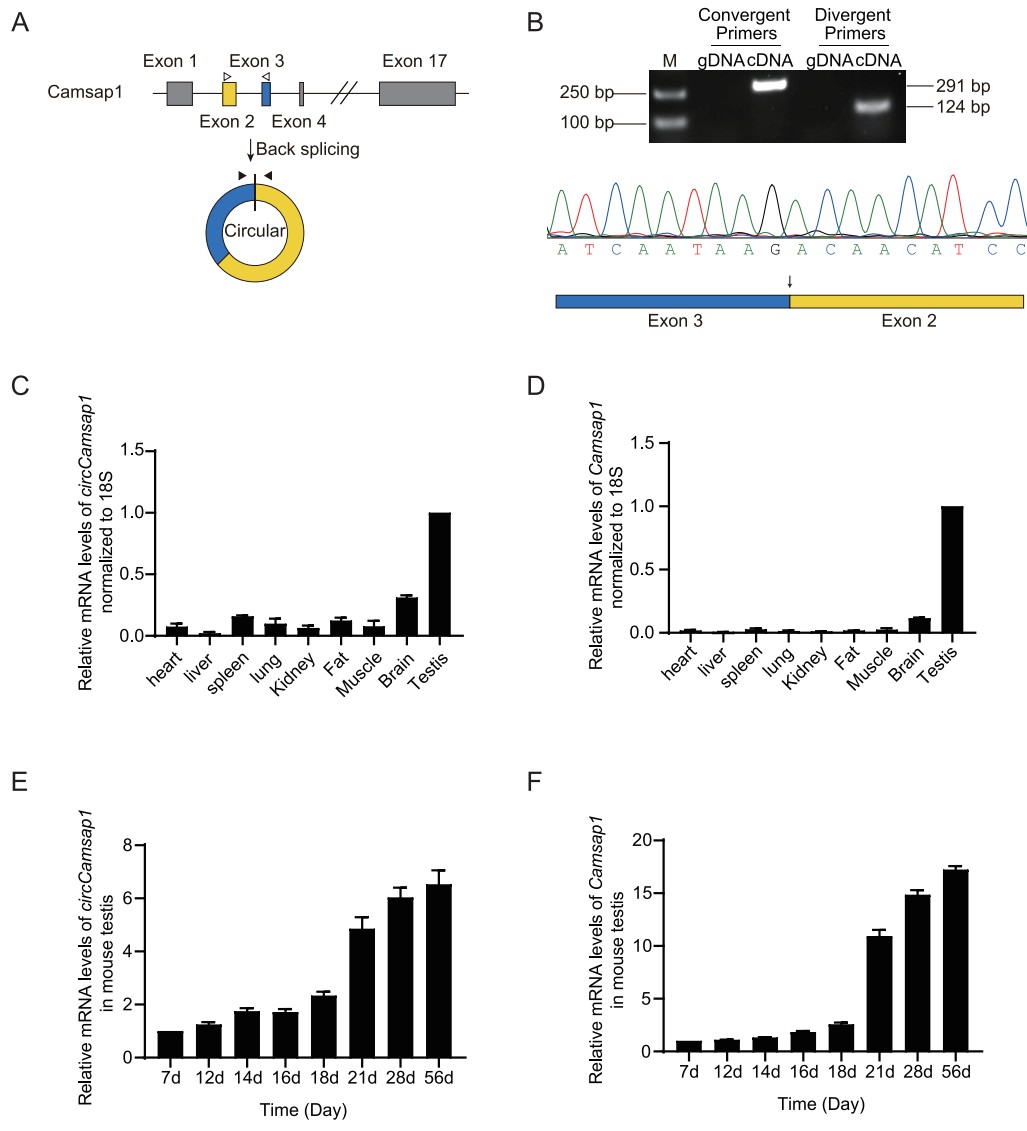

**Figure 1** The expression of *circCamsap1* in mice. (A) The annotated genomic region of *Camsap1* is described, and the head-to-tail splicing form of *circCamsap1* along with the validation strategy are shown. Solid arrows represent divergent primers used to detect *circCamsap1*. Hollow arrows represent convergent primers used to detect *Camsap1*. (B) The convergent primers detected *Camsap1* only in mouse testicular cDNA. The divergent primers detected *circCamsap1* in mouse testicular cDNA, but not in gDNA. Sanger sequencing of the PCR products confirmed the presence of *circCamsap1* in mouse testes. (C) RT-PCR was used to detect the expression levels of *circCamsap1* in various tissues of mice. Expression levels were normalized to 18S ($n = 3$). (D) RT-PCR was used to detect the expression levels of *Camsap1* in various tissues of mice. Expression levels were normalized to 18S ($n = 3$). (E) Relative expression levels of *circCamsap1* in mouse testes at different ages (in days) ($n = 3$). (F) Relative expression levels of *Camsap1* in mouse testes at different ages (in days) ($n = 3$).

## Generation of *circCamsap1*⁻ᐟ⁻ mice

Subsequently, we utilized CRISPR/Cas9 technology to generate *circCamsap1* knockout mice. Recognizing the significance of flanking complementary sequences in introns for circRNA formation (*Zhang et al., 2014*), we designed two sgRNAs targeting gene sites

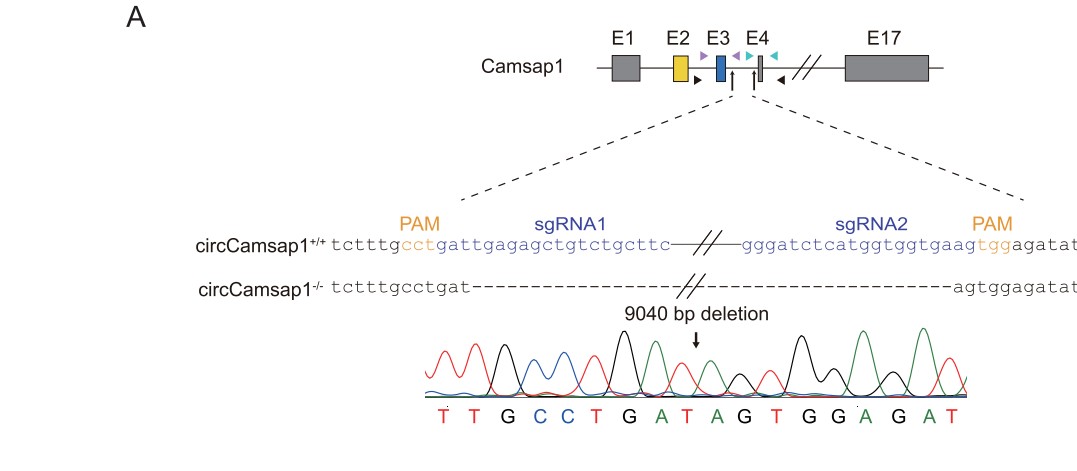

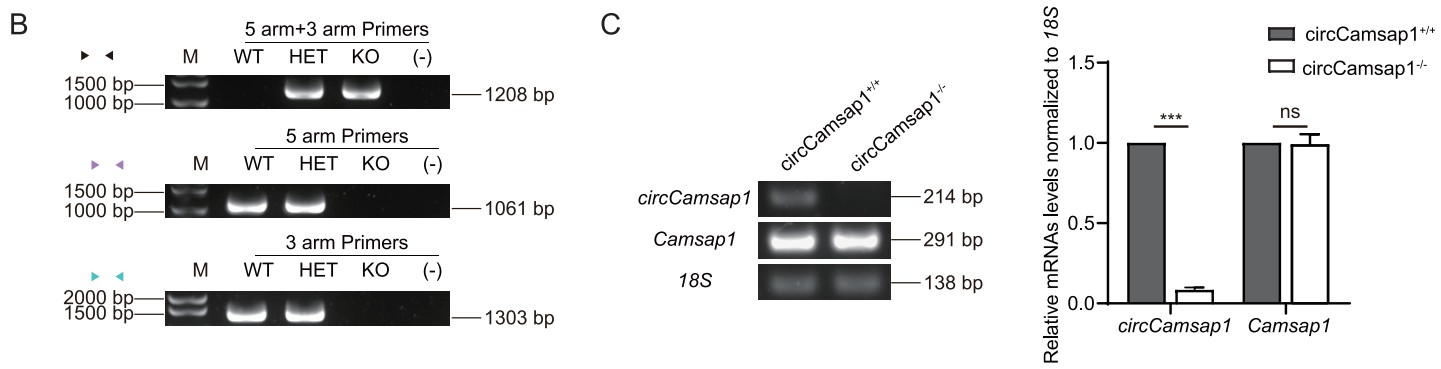

**Figure 2 The generation of *circCamsap1*⁻/⁻ mice.** (A) Schematic strategy of *circCamsap1*⁻/⁻ mice generated by CRISPR/Cas9 technology. Two sgRNAs were designed to target the intron within exons 3 and 4 of *Camsap1* for knockout. Different colored arrows represent various identification primers, and Sanger sequencing results confirmed the successful deletion of 9,040 bp from the *Camsap1* gene. (B) Identification of *circCamsap1*⁻/⁻ mice by PCR. (C) RT-PCR analysis was conducted to assess *circCamsap1* and *Camsap1* mRNA levels in *circCamsap1*⁺/⁺ and *circCamsap1*⁻/⁻ mice. The statistical graph shows the gray values after three repetitions (*n* = 3). All data are means ± SEM, tested using a two-tailed unpaired t-test, ***P < 0.001, ns indicates no difference.

within the intron situated between exon 3 and exon 4 of *Camsap1* (Fig. 2A). Genotype analysis, conducted through PCR and then Sanger sequencing, demonstrated the successful deletion of a 9,040 bp intron sequence crucial for *circCamsap1* formation (Figs. 2A and 2B). To ensure the specific knockout of *circCamsap1* without compromising the expression of its parental gene *Camsap1*, we conducted an RT-PCR experiment. The results validated the deletion of *circCamsap1* while preserving the expression of *Camsap1* (Fig. 2C).

## *CircCamsap1*⁻/⁻ mice are fertile with normal spermatogenesis

*CircCamsap1*⁺/⁺ males exhibited normal fertility (Fig. 3A). We compared the testicular/body weight ratio of *circCamsap1*⁻/⁻ mice to *circCamsap1*⁺/⁺ mice and found no obvious difference (Figs. 3B–3D). Meanwhile, the testes and epididymides of *circCamsap1*⁻/⁻ mice were normal in size compared with *circCamsap1*⁺/⁺ mice (Fig. 3E). In addition, we performed H&E staining on the testes, revealing normal spermatogenic cells of all stages of adult (postnatal day 56) *circCamsap1*⁻/⁻ mice (Fig. 3F). In order to analyze the

spermatocytes of $circCamsap1^{-/-}$ mice, we examined the localization of the synaptonemal complex SYCP3 and γ-H2AX, which mark the formation and repair of meiotic DNA double-strand breaks, in the surface-spreading spermatocyte nuclei (*Zickler & Kleckner, 1999*; *Hunter et al., 2001*). The various stages of meiotic prophase were determined by immunofluorescence staining. We found that, compared with $circCamsap1^{+/+}$ mice, the meiotic prophase cells of the $circCamsap1^{-/-}$ mice also completely contained the typical four different cytological stages, including leptotene, zygotene, pachytene and diplotene (Fig. 3G). Subsequently, we quantified cells in each of the four stages of meiotic prophase, and the results demonstrated no significant differences between $circCamsap1^{-/-}$ mice and $circCamsap1^{+/+}$ mice (Fig. 3H). Therefore, we concluded that the absence of circCamsap1 does not affect meiosis and spermatogenesis in mice.

### *CircCamsap1$^{-/-}$* mice have normal sperm parameters

By conducting histological analysis on the mouse cauda epididymis, we observed that the sperm density in the $circCamsap1^{-/-}$ mice appeared to be normal compared to that of the $circCamsap1^{+/+}$ mice (Fig. 4A). No notable distinctions were found in spermatozoa morphology from the epididymal cauda between $circCamsap1^{+/+}$ and $circCamsap1^{-/-}$ mice (Fig. 4B). Additionally, we utilized the CASA system to measure sperm parameters, including count, motility, and progressive motility in mice. The results showed that there was no significant difference in these parameters between $circCamsap1^{-/-}$ and $circCamsap1^{+/+}$ mice (Figs. 4C and 4D).

## DISCUSSION

Here, we constructed a *circCamsap1* knockout mouse using CRISPR/Cas9 technology without affecting the expression of the parental gene, *Camsap1*. We observed that in male mice lacking *circCamsap1*, fertility, testicular tubule structure, progression of meiosis, as well as sperm motility and characteristics were all completely normal compared to wild-type mice. This indicates that *circCamsap1* is not essential for male fertility.

Due to the sequence overlaps between circRNAs and their parental gene, it is challenging to specifically knock out circRNAs without affecting the parental gene (*Li, Yang & Chen, 2018*). Consequently, the investigation of the biological significance of individual circRNAs has encountered some obstacles. To date, only a few circRNA knockout models have been established in mice. A scientific research team first reported that deletion of *circPan3* in intestinal stem cells (ISCs), impaired the immune cell-mediated self-renewal of Lgr5[+] ISCs and epithelial regeneration in C57BL/6 mice (*Zhu et al., 2019*). Subsequently, their team reported that knockout of *circKnct2* resulted in intestinal ILC3 activation and severe colitis in mice (*Liu et al., 2020*). However, only a very small number of circRNA knockout mouse models have been used to study spermatogenesis (*Gao et al., 2020*; *Zhang et al., 2023*). Here, we designed sgRNAs targeting the circularization element and successfully constructed circRNA knockout mice, demonstrating the feasibility of this knockout approach. This provides valuable assistance for future *in vivo* studies on the biological functions of individual circRNAs.

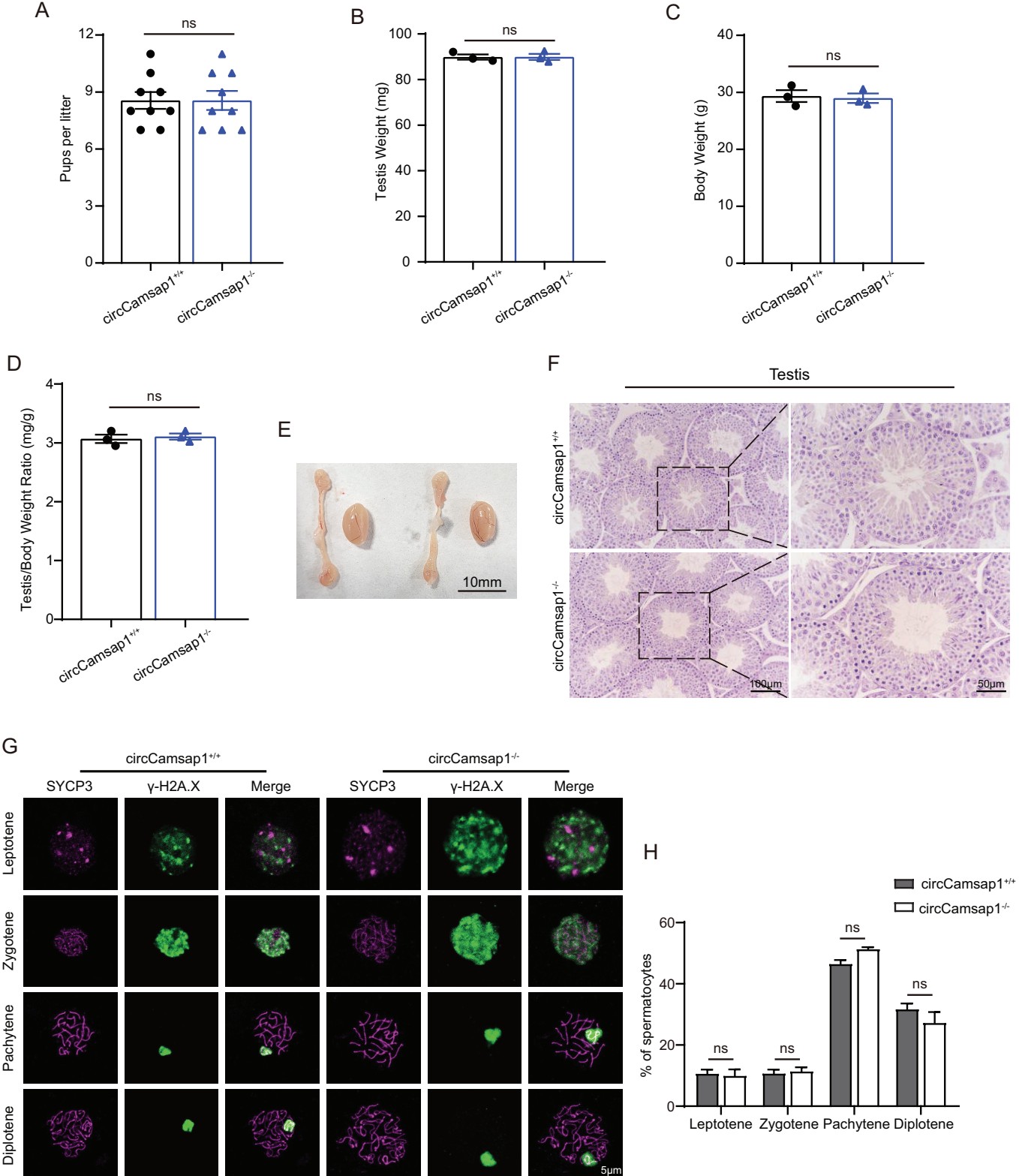

**Figure 3** *circCamsap1⁻/⁻* **mice have normal spermatogenic and meiotic processes.** (A) The litter size of *circCamsap1*⁺/⁺ and *circCamsap1*⁻/⁻ mice are shown. (B-D) Testis weight, body weight and testis/body weight ratio of adult (postnatal day 56) *circCamsap1*⁺/⁺ and *circCamsap1*⁻/⁻ mice. (E) Representative images of testes and epididymides of adult (postnatal day 56) *circCamsap1*⁺/⁺ and *circCamsap1*⁻/⁻ mice. (F) H&E staining of the

**Figure 3 (continued)**
testis from adult (postnatal day 56) *circCamsap1$^{+/+}$* and *circCamsap1$^{-/-}$* mice. (G) Immunofluorescence staining using antibodies against SYCP3 and γ-H2AX in surface-spreading spermatocytes of both *circCamsap1$^{+/+}$* and *circCamsap1$^{-/-}$* mice. (H) Meiotic stage frequencies in *circCamsap1$^{+/+}$* and *circCamsap1$^{-/-}$* testes (n = 3). All data are means ± SEM, tested using a two-tailed unpaired t-test, ns indicates no difference.

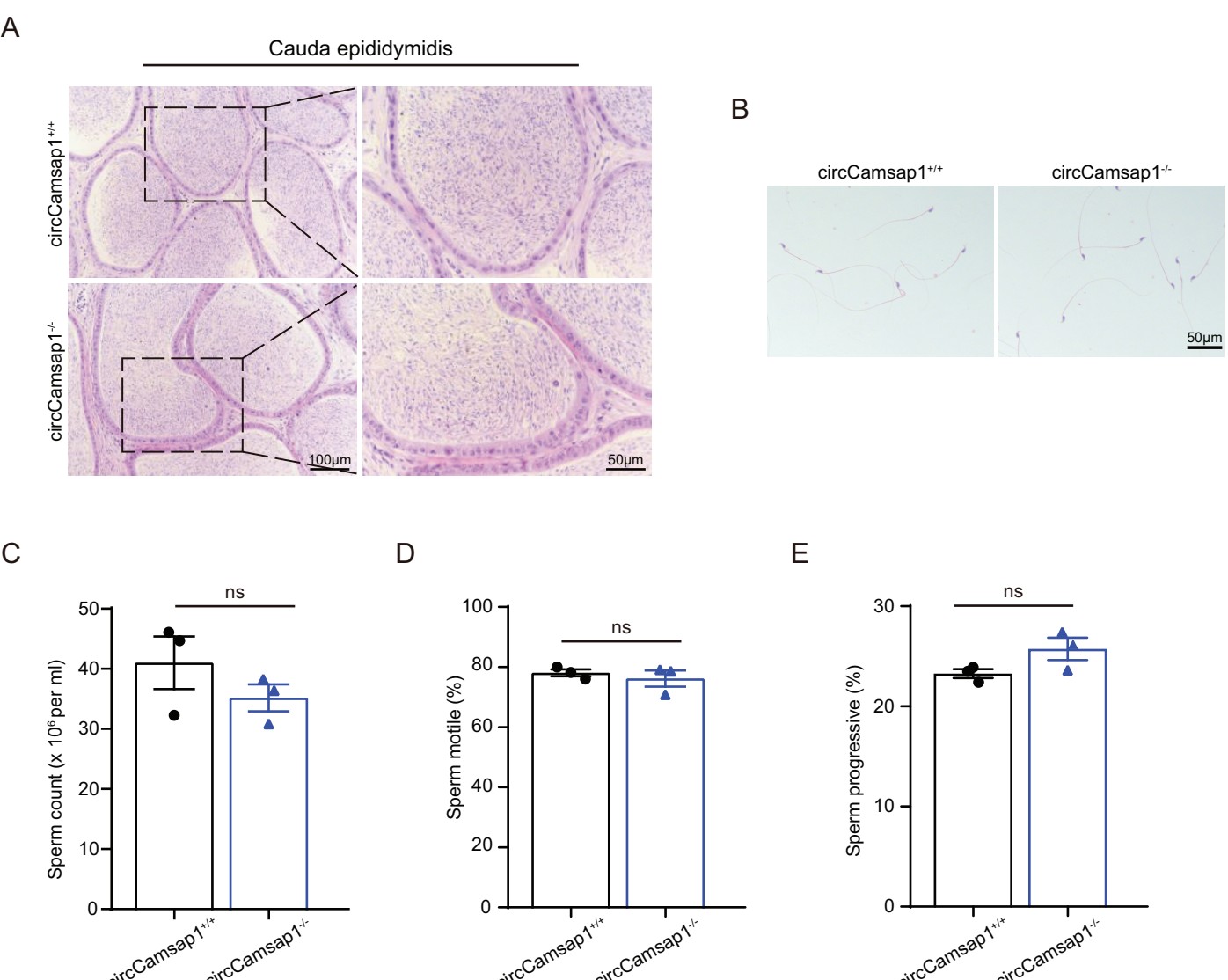

**Figure 4 Spermatozoa appear normal in *circCamsap1$^{-/-}$* mice.** (A) H&E staining of cauda epididymidis from adult (postnatal day 56) *circCamsap1$^{+/+}$* and *circCamsap1$^{-/-}$* mice. (B) H&E staining of sperm from adult (postnatal day 56) *circCamsap1$^{+/+}$* and *circCamsap1$^{-/-}$* mice. (C–E) Sperm count, sperm motile and sperm progressive were assessed in adult (postnatal day 56) *circCamsap1$^{+/+}$* and *circCamsap1$^{-/-}$* mice (n = 3). All data are presented as means ± SEM, analyzed using a two-tailed unpaired t-test, ns indicates no difference.

Here, we investigated whether circCamsap1, a circRNA relatively highly expressed in the testes, affects spermatogenesis. Previous studies have shown that loss of Camsap1 results in mice displaying abnormalities in sperm morphology, reduced sperm count,

decreased sperm motility, ultimately leading to male infertility (*Hu et al., 2023*). However, upon examining fertility and the spermatogenesis process in male circCamsap1 knockout mice, we did not observe any reproductive defects.This suggests that circCamsap1 is not essential for male fertility and indicates that while certain genes play significant roles in male reproduction, their derived circRNAs may not have equally important functions. Studying the role of circRNAs in reproductive biology requires more consideration of their own functions and effects, rather than solely focusing on those of their parental genes.

CircRNAs are highly expressed in testicular tissue, second only to the brain, suggesting their important role in spermatogenesis (*You et al., 2015*). The study found abundant circRNAs in mouse spermatogonial stem cells, spermatogonia, preleptotene spermatocytes, pachytene spermatocytes, round spermatids and elongating spermatids (*Lin et al., 2016*; *Tang et al., 2020*). Most of the circRNAs expressed during mouse spermatogenesis are derived from exonic regions (*Wu et al., 2019*), and some of them originate from genes that are specifically expressed in the testes, indicating that these circular RNAs may have important functions (*Glazar, Papavasileiou & Rajewsky, 2014*). However, research on the function of circRNAs in spermatogenesis is still in early stages. In this study, we investigated the role of *circCamsap1* in spermatogenesis *in vivo* and demonstrated that it is not an essential gene for male reproduction. However, one possible explanation may lie in the existence of compensatory mechanisms within the biological system, where other genes or regulatory pathways may have compensated for the loss of circCamsap1, thereby mitigating any anticipated impact on reproductive outcomes. Additionally, while we made efforts to comprehensively assess various aspects of reproductive function, including fertility parameters and spermatogenesis outcomes, the sample size of three mice may still be insufficient. Furthermore, our observations were limited to male mice only. Despite these limitations, our study provides valuable insights into the role of circCamsap1 in male reproduction and emphasizes avenues for further research to address these limitations and deepen our understanding of circRNA function in reproductive biology.

## CONCLUSIONS

In summary, we successfully generated *circCamsap1$^{-/-}$* mice with a 9,040 bp deletion using CRISPR-Cas9 gene editing technology. Our results provide evidence that despite the relatively high expression of *circCamsap1* in mouse testes, the deletion of this gene does not impact the reproductive process in male mice.

## ACKNOWLEDGEMENTS

We thank Jinyang Cai for continuous support with microscopy.

### Funding

The study was supported by the Science and Technology Project of Changzhou Health Commission (QN202317), the Scientific Research Project of Changzhou Medical Center of

Nanjing Medical University (CMCP202303 to HL), the Jiangsu Funding Program for Excellent Postdoctoral Talent (2023ZB319 to HL), and the Changzhou Sci&Tech Program (Grant No. CJ20230073) The funders had no role in study design, data collection and analysis, decision to publish, or preparation of the manuscript.

## Grant Disclosures

The following grant information was disclosed by the authors:
Science and Technology Project of Changzhou Health Commission: QN202317.
Scientific Research Project of Changzhou Medical Center of Nanjing Medical University: CMCP202303 to HL.
Jiangsu Funding Program for Excellent Postdoctoral Talent: 2023ZB319 to HL.
Changzhou Sci&Tech Program: CJ20230073.

## Competing Interests

The authors declare that they have no competing interests.

## Author Contributions

- Shu Zhang conceived and designed the experiments, performed the experiments, analyzed the data, prepared figures and/or tables, authored or reviewed drafts of the article, and approved the final draft.
- Haojie Li conceived and designed the experiments, performed the experiments, analyzed the data, prepared figures and/or tables, authored or reviewed drafts of the article, and approved the final draft.
- Wei Jiang performed the experiments, analyzed the data, prepared figures and/or tables, and approved the final draft.
- Xia Chen analyzed the data, prepared figures and/or tables, and approved the final draft.
- Han Zhou analyzed the data, prepared figures and/or tables, and approved the final draft.
- Chang Wang analyzed the data, prepared figures and/or tables, and approved the final draft.
- Hao Kong analyzed the data, prepared figures and/or tables, and approved the final draft.
- Yichao Shi conceived and designed the experiments, analyzed the data, authored or reviewed drafts of the article, and approved the final draft.
- Xiaodan Shi conceived and designed the experiments, analyzed the data, authored or reviewed drafts of the article, and approved the final draft.

## Animal Ethics

The following information was supplied relating to ethical approvals (*i.e.*, approving body and any reference numbers):

Institutional Animal Care and Use Committee of Nanjing Medical University provided full approval for this research (Approval No. IACUC-2307030).

## Data Availability

The raw data are available in the Supplemental Files.

## Supplemental Information

Supplemental information for this article can be found online at http://dx.doi.org/10.7717/peerj.17399#supplemental-information.

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
