# Peer review of "CircCamsap1 is dispensable for male fertility in mice"

_PeerJ, doi:10.7717/peerj.17399_

## Round 0.1 · original submission · Major Revisions

Please address issues pointed out by both reviewers and amend the manuscript accordingly.

**Language Note:** The review process has identified that the English language must be improved. PeerJ can provide language editing services - please contact us at [email protected] for pricing (be sure to provide your manuscript number and title). Alternatively, you should make your own arrangements to improve the language quality and provide details in your response letter. – PeerJ Staff

Reviewer 1 ·

Basic reporting

no comment

Experimental design

no comment

Validity of the findings

no comment

Additional comments

Abstract
The abstract effectively outlines the methods and results of the study, some points could be critiqued, and some grammar errors too.
1.The abstract concludes that circCamsap1 may not play an essential role in physiological spermatogenesis based on the findings in circCamsap1 knockout mice. However, it does not discuss the potential implications of this conclusion for the broader understanding of circRNA function in male reproduction. Providing a brief discussion of the potential impact of the findings would enhance the abstract.
2.The abstract states that the results "offer guidance to reproductive biologists, directing their attention toward genes crucial for fertility and helping avoid redundant research efforts." While this statement is valuable, it could be strengthened by providing specific insights or recommendations based on the study's findings.
3.The abstract could benefit from clearer and more precise language to convey the specific findings and implications of the study. For example, it could provide more detailed information about the fertility and sperm motility assessments and their implications for understanding circCamsap1 function in male reproduction.
Introduction (check some grammar errors)
1.The introduction could benefit from a clearer emphasis on the novelty and originality of the study. For instance, it should explicitly state how the study fills a critical knowledge gap and advances the current understanding of circRNAs in spermatogenesis. This could be achieved by highlighting specific gaps in the current literature and explaining how the study's findings will address these gaps.
2.The introduction could benefit from a clearer emphasis on the novelty and originality of the study. For instance, it should explicitly state how the study fills a critical knowledge gap and advances the current understanding of circRNAs in spermatogenesis. This could be achieved by highlighting specific gaps in the current literature and explaining how the study's findings will address these gaps.

Materials and methods
The methodology appears well-described and appropriate for the study objectives
Results and Discussions
Looks nice
Conclusions
The conclusion lacks information about the potential function or significance of circCamsap1 in the reproductive process. Without a discussion of the expected impact of circCamsap1, it is challenging to assess the biological relevance of the deletion.
Here are some recommendations during the revision of your conclusion
1.Clarify the expected biological impact of circCamsap1 in the reproductive process to provide context for the deletion study.
2.Discuss unexpected findings or provide more details on the lack of impact, considering potential compensatory mechanisms or limitations.
3.Incorporate statistical measures to support the claim that the deletion does not impact the reproductive process.
4.Since the study is specifically focused on male mice, explicitly state this in the conclusion.
5.Acknowledge Limitations: Highlight any limitations of the study, such as sample size or specific aspects of reproductive function assessed.

Cite this review as

·

Basic reporting

no comment. See detailed information below.

Experimental design

no comment. See detailed information below.

Validity of the findings

no comment. See detailed information below.

Additional comments

As a reviewer with a background in reproductive research, I have meticulously examined the manuscript titled "CircCamsap1 is dispensable for male fertility in mice" (#92652). Despite its presentation of negative results, the study potentially offers intriguing insights for future research. However, there are several areas requiring attention prior to publication:
1. The manuscript contains sections of imprecise language use and would benefit from revision by a professional English editing service to ensure clarity and correctness.
2. It is advised to provide the full designation of Camsap1 at its initial mention to aid reader comprehension.
3. The Introduction should better highlight the novelty and scope of the current study to clearly articulate its contribution to the existing body of knowledge.
4. The manuscript must specify the ages of the mice used. The results indicated an increase in circCamsap1 after 21 days; thus, different knockout times could yield varying impacts on fertility. For instance, a knockout at postnatal day (PND) 5 might hinder the initial spermatogenic wave.
5. The Methods section lacks essential details. It should specify the duration of the fertility assessment or an endpoint, for example, whether it spans two nights (48 hours) or just one (12 hours). Additionally, clarification is needed on what is considered 'adult' age for mice—is it PND 28, PND 56, or another age?
6. A statistical chart should accompany Figure 2C to better present and interpret the data.
7. The rationale behind having nine data points in Figure 3A with only three mice per group needs clarification. Does this imply each mouse was mated with three females? Such critical information must be detailed in the Methods section.
8. Considering recent research positing CAMSAP1 as indispensable for spermatogenesis, it is crucial to discuss how the current findings align or contrast with these earlier studies, thereby situating the present work within the broader research landscape.
9. The manuscript does not adequately address its limitations, a critical aspect for studies reporting negative results. It is essential to discuss the potential implications and limitations of these findings to provide a balanced and comprehensive view.

---

## Round 0.2 · accepted · Accept

Concerns of the reviewers were addressed and revised manuscript is acceptable now.

·

Basic reporting

no comment

Experimental design

no comment

Validity of the findings

no comment

Additional comments

no comment